CC ()

2

# 1 Risk assessment of meteorological drought in China under RCP

## scenarios from 2016 to 2050

3 Kuo Li Jie Pan 4 (Institute of Environment and Sustainable Development in Agriculture, CAAS, Beijing, China) 5 Abstract: Climate change has been a hotspot of scientific research in the world for decades, 6 which caused serious effects of agriculture, water resources, ecosystem, environment, human 7 health and so on. In China, drought accounts for almost 50% of the total loss among all the 8 meteorological disasters. In this article the interpolated and corrected precipitation of one GCM 9 (HadGEM2-ES) output under four emission scenarios (RCP2.6, 4.5, 6.0, 8.5) were used to analyze 10 the drought. The standardized precipitation index (SPI) calculated with these data was used to assess the climate change impact on droughts from meteorological perspectives. Based on five 11 12 levels of SPI, an integrated index of drought hazard (IIDH) was established, which could explain 13 the frequency and intensity of meteorological drought in different regions. According to 14 yearbooks of different provinces, 15 factors have been chosen which could represent the impact 15 of drought on human being, crops, water resources and economy. Exposure index, sensitivity 16 index and adaptation index have been calculated in almost 2400 counties and vulnerability of 17 drought has been evaluated. Based on hazard and vulnerability evaluation of drought, risk assessment of drought in China under the RCP2.6, 4.5, 6.0, 8.5 emission scenarios from 2016 to 18 19 2050 has been done. Results from such a comprehensive study over the whole country could be 20 used not only to inform on potential impacts for specific sectors but also can be used to 21 coordinate adaptation/mitigation strategies among different sectors/regions by the central 22 government.

Keywords: Risk assessment, Climate change, Meteorological drought, RCP scenarios,
 Vulnerability evaluation, China

# **1 Introduction**

According to IPCC 5<sup>th</sup>, it has been successively warmer in the past 30 years on the Earth's surface 26 than any decade since 1850 (IPCC, 2013). Climate change has been a hotspot of scientific 27 28 research in the world for decades, which caused serious effects of agriculture, environment, 29 ecosystem, human health and so on. In China, the temperature has increased remarkably since 30 1850 (Wang and Chen, 2014; Wei and Chen, 2011). The variations of precipitation in different 31 regions of China for the past hundred years are different and distinguishing, for example, North 32 and Northeast districts of China becoming drier, East and Central districts of China becoming 33 moister (Liu et al., 2011; Ma et al., 2012; Wu et al., 2012). Due to climate change, the disasters related climate become much more serious, which lead to huge losses of lives and economy, 34 35 destructions of ecology and environment, damage of human health and so on. According to statistical data, drought accounts for almost 50% of the total loss among all the meteorological 36 37 disasters in China (Liu, 2012). With the variation of precipitation distribution and rising 38 temperature, the intensity and frequency of drought disasters are changing quickly in the whole world (Prudhomme et al., 2014; Burke et al., 2006; Trenberth et al., 2014; Dai, 2012). So it is 39

important to do research on the trend of spatial and temporal distribution of drought under 41 climate change in China.

Drought is a complex and natural phenomenon mainly caused by low rainfall in a constant 43 period which is characterized by several properties such as frequency, intensity and duration 44 (Mishra and Singh, 2010; Wilhite, 2000; Van Loon and Van Lanen, 2012). According to the impact, 45 droughts can be classified into different forms such as meteorological drought, agricultural drought, hydrological drought and socio-economical drought (Tallaksen and Lanen, 2005; Hayes 46 47 et al., 2007; Van Loon and Laaha G., 2015). Meteorological drought is characterized by lack of 48 precipitation over an extended period; hydrological drought is characterized by persistent 49 reduction in runoff; agricultural drought is characterized by reduction of soil moisture and crops yield (Hisdal and Tallaksen, 2003; Keyantash and Dracup, 2002; Sheffield and Wood, 2008). Along 50 51 with the reduction of precipitation, runoff and soil moisture, the shortage of water supply for 52 population, livestock, industry, ecology, environment may become much more serious, then 53 socio-economical drought would break out. Meteorological drought is mainly determined by 54 climatic conditions and atmospheric circulations. However, the other three types of drought 55 (agricultural drought, hydrological drought and socio-economical drought) are primarily 56 influenced by both natural and anthropogenic systems, especially socio-economical drought 57 which is affected mostly by human activities (Wilhite, 2000; Wisser et al., 2010). Usually 58 meteorological drought is the base of the other three drought types, so it is important to do 59 research on meteorological drought in large scale which would reveal the potential trends of 60 drought disasters.

A lot of studies are performed to analyze the frequency, intensity and duration of droughts by 62 using different indexes, such as the Standardized Precipitation Index (SPI) (McKee et al., 1993), 63 the Standard Runoff Index (SRI) (Shukla and Wood, 2008), the Palmer Drought Severity Index 64 (PDSI) (Palmer, 1965; Wells et al., 2004), the Standardized Precipitation-Evapotranspiration Index 65 (SPEI) (Vicente-Serrano et al., 2010) and the Supply-Demand Drought Index (SDDI) (Rind et al., 1990). Comparing the different indexes, SPI and PDSI are appropriate for meteorological drought; 66 67 SRI is proper for hydrological drought; SPEI is good for agricultural drought; SDDI is much better for socio-economical drought. For this research, SPI is chosen to establish integrated drought 68 69 index, which would evaluate the hazard of drought under the RCP (Representative Concentration 70 Pathways) emission scenarios (Van Vuuren et al., 2011a; 2011b) from 2016 to 2050.

China has been affected frequently by drought in the past thousand years (Zou et al., 2005; 72 Dai, 2012; Dai et al., 2004; Ma and Fu, 2003). According to the Ministry of Water Resources of 73 China (MWRC, 2011), drought disasters have caused huge yield loss, nearly 39.2 billion kilograms 74 annually. A lot of researches have focused on drought trends and impact of drought under 75 climate change (Xu et al., 2012; Chen et al., 2006; Zhou et al., 2009; Qian et al., 2014), in which 76 changes on scope, intensity, duration and frequency of drought in China at a nationwide scale 77 have been explored (Zhou et al., 2006; Yuan et al., 2012; Chen et al., 2013; Nath et al., 2014). 78 According to Wang et al. (2003) and Wang et al. (2011), the drought affected more and more 79 areas in which severe droughts became much more frequent over the past 60 years, so risk 80 assessment on drought disasters should be carried out as soon as possible. Wu et al. (2011) 81 revealed that almost 30% of the total farmland in China is vulnerable to drought. Zhang et al. 82 (2013a) explored the frequency of extreme drought and analyzed the changes of geographic 83 distributions from 1960 to 2009 in Southwest China. There are a few studies which have analyzed

the variations of volume and spatial distribution of water resources under climate change in 85 North China, South China and the whole country (Leng et al., 2015; Xu et al., 2009c; Li et al., 2010; Jiang et al., 2007; Qiu, 2010; Yang et al., 2012; Wang et al., 2012; Guo et al., 2002; Wang et al., 86 87 2014). However, very few studies have assessed the risk of drought under climate change, 88 especially the vulnerability of drought across the whole country. It is obvious that most 89 researchers pay attention to the natural characteristics of drought, such as scope, intensity, duration, frequency and so on. But the social features of drought are not concerned enough, 90 91 which is mainly about the hazard-affected bodies, like population, agriculture, industry, cities, 92 water and so on. It is difficult to evaluate the impact of drought on these bodies, especially under 93 climate change drought may become much more intensive and frequent. Risk assessment provides us a good method to evaluate the hazard and vulnerability of drought, which could give 94 95 us a clear picture of drought distribution and enable more effective drought management plans 96 to be developed.

In this research, a coupled Earth System Model - HadGEM2-ES (Collins et al., 2008) has been 98 used to generate the precipitation under the RCP2.6, 4.5, 6.0, 8.5 emission scenarios in the future. 99 The standardized precipitation index (SPI) was used to establish an integrated index of drought 100 hazard (IIDH), which could explore the frequency and intensity of meteorological drought in 101 different regions. According to yearbooks of different provinces, exposure index, sensitivity index 102 and adaptation index have been calculated in almost 2400 counties and vulnerability of drought has been evaluated. Based on hazard and vulnerability evaluation of drought, risk assessment of 103 104 drought in China under the RCP2.6, 4.5, 6.0, 8.5 emission scenarios from 2016 to 2050 has been 105 done. Obviously, results from such a comprehensive study over the whole country could be used 106 not only to inform on potential impacts for specific sectors but also can be used to coordinate 107 adaptation/mitigation strategies among different sectors/regions by the central government.

#### 2 Data and methods 108

#### 2.1 Data 109

The study area includes the whole mainland China except Taiwan islands because of the unavailability of data from Taiwan. As mentioned above, risk assessment includes two 111 aspects-hazard evaluation and vulnerability evaluation. The data should be collected from the 112 113 two aspects. On one hand, climate scenarios data (precipitation) are needed for drought hazard evaluation. On the other hand, the data about society, economy, population, water resources, 114 115 forest and so on in 2373 counties of whole China should be collected and prepared. The vulnerability evaluation of drought is complicated and data sources are various, so it is necessary 116 to carry out reliability test and preprocess the historical and observed data to avoid the distortion 117 118 of double counting.

In this study, the projected daily precipitation from GCM HadGEM-ES is the simulation result 120 of 1951-2099 under RCP scenarios which is interpolated and corrected. HaGEM-ES (Hadley 121 <u>G</u>lobal <u>Environment M</u>odel <u>2</u> - <u>Earth System</u>) is designed to run the major scenarios for IPCC AR5 by the UK Met Office Hadley Centre for CMIP5 (The World Climate Research Programme's 122 Coupled Model Intercomparison Project phase 5) centennial simulations. The horizontal 123

resolution of HadGEM2-ES Model's raw output is 1.875°×1.25°. The ISI-MIP (The Inter-Sectorial 125 Impact Model Intercomparison Project) changed the data to 0.5°×0.5° at horizontal resolution with the bilinear interpolation method, and a statistical bias correction algorithm based on 126 127 probability distribution is used to correct the interpolation result (Piani et al., 2010; Hagemann et 128 al., 2011).

On the other hand, the data for vulnerability evaluation of drought is collected from 129 130 social-economic Yearbooks of different provinces, water resources bulletins, forest resources 131 bulletins and so on. In the study, counties and districts are set as standard statistical units, which 132 are used for analysis and calculation of exposure factors, sensitivity factors and adaptive capacity 133 factors. There are so many factors which could affect the vulnerability of drought. So it is important for us to choose proper and effective factors in China. Based on the relationship 134 135 between drought disasters and hazard-affected bodies, the most important factors are selected 136 which could be measurable and comparable. Due to the complexity and diversity of drought 137 vulnerability factors of which the units are different, all the evaluation factors are normalized into 138 non-dimensional by geometric average processing in order to be convenient for utilization.

#### 2.2 Drought Indices 139

There are several drought indices created to identify the different types of drought disasters. 141 According to McKee et al. (1993), the standardized precipitation index (SPI) is designed to quantify the precipitation deficit for multiple time scales, which could assess the impact of 142 143 climate change on drought disasters from meteorological perspectives. SPIs in different time 144 scales reflect the degrees of shortage on water resources due to meteorological drought. The 145 changes of ground water, streamflow, underwater and reservoir storage are closely related to the 146 precipitation anomalies in a long term, but soil moisture conditions are connected with the precipitation anomalies in a short term. SPI was testified to be effective to explore the intensity 147 of meteorological drought. It is the most important drought indice to reveal the potential drought 148 149 trends, which is the basis of judging agricultural drought, hydrological drought and 150 socio-economical drought. For these reasons, the SPIs for 3-month, 6-month, 12-month, 151 24-month, and 48-month time scales are originally calculated in this article, which are based on 152 the precipitation records in different time scales. According to Edwards and McKee (1997), the long-term record is fitted to a probability distribution, which is then transformed into a normal 153 154 distribution so that the mean SPI for the location and desired period is zero. Positive SPI values indicate it is greater than median precipitation, while negative values indicate it is less than 155 156 median precipitation. Because the SPI is normalized, wetter and drier climates situations can be 157 represented in the same way.

Drought intensities reflected from the SPI are defined by the classification system shown in 158 159 the SPI Values table. According to the define, a meteorological drought event would occur if the 160 SPI is continuously negative and reaches the critical value that the SPI is -1.0 or less. The drought 161 event would end when the SPI becomes positive. Each drought event has a duration defined by 162 its beginning and end. The drought intensity is the positive sum of the SPIs within a drought event, which is to accumulate the magnitudes of drought for all the duration. According to 163 164 different cases, SPI could be classified into mild drought, moderate drought, severe drought and 165 extreme drought. Based on the standardized SPIs, the rarity of meteorological drought is

166 determined by the probability of the precipitation during the duration of drought (Guttman, 1998; 167 Kogan, 1995; Wilhite and Glantz, 1985).

| Tab 1 Drought   | grades and we | ighting factor | according to SPI |
|-----------------|---------------|----------------|------------------|
| iab. I Diougiit | glaues and we | ignuing factor | according to SFT |

| Degree of drought | Value of SPI                                   | Weighting factor |  |
|-------------------|------------------------------------------------|------------------|--|
| Mild drought      | -1.0 <spi≤-0.5< td=""><td>0.1</td></spi≤-0.5<> | 0.1              |  |
| Moderate drought  | -1.5 <spi≤-1.0< td=""><td>0.2</td></spi≤-1.0<> | 0.2              |  |
| Heavy drought     | -2.0 <spi≤-1.5< td=""><td>0.3</td></spi≤-1.5<> | 0.3              |  |
| Excessive drought | SPI≪-2.0                                       | 0.4              |  |

Firstly, the shape parameter and scale parameter (Yuan and Zhou, 2004) at each grid of the 169 whole study area are estimated with the corrected precipitation from HadGEM2-ES in 1971-2000 170 171 according to maximum likelihood estimation (MLE). With the above parameters, the SPI is calculated for 12-month time scale from 2016-2050 at each grid. Secondly, the degree of drought 172 173 intensity is graded according to the value of SPI, and each level is given a particular weighting factor, shown in Table 1. Then, for each grid, the frequencies of different drought (from mild to 174 175 excessive drought) are counted. Finally, these frequencies are calculated by using the corresponding weighting factors in Table 1 to produce a new dataset, which covers the integrated 176 177 indexes of drought hazard.

So, based on the four levels of SPI, an integrated index of drought hazard (IIDH) was 179 established, which could explain the frequency and intensity of meteorological drought in 180 different regions.

#### 2.3 Vulnerability evaluation 181

According to IPCC (2013), vulnerability is defined as the propensity or predisposition to be adversely affected in IPCC 5<sup>th</sup> report, which is not always corresponding definition in numerous 183 literatures (Houghton et al., 2001; Cannon, 1994; Cutter, 1996a). But the connotation of 184 185 vulnerability is becoming much clear, which is mainly about the inherent characteristics of acceptors (human being, society, economy, agriculture, water et al.) when they are faced with 186 187 different coerces or threats, such as climate change, extreme events, disasters and so on. Under 188 climate change, there are a lot of changes on drought trends in China. In some regions, it becomes much more serious and frequency; in other regions, it becomes weakening and 189 declining. But the economic and environmental losses caused by drought disasters are becoming 190 191 much more tremendous. So it is important to evaluate the impacts of climate change on drought. 192 Vulnerability assessment could reveal the relationship between stress factors and acceptors, 193 distribution of vulnerable areas, and degree of vulnerability. With the increasing knowledge on 194 vulnerability, the vulnerability evaluation model has become stable and clear, which contains 195 three aspects: exposure, sensitivity and adaptive capacity. Most of researchers have accepted the 196 vulnerability evaluation model. In this research, exposure is the extent to which the acceptors are 197 subject to potential drought. Sensitivity is the reaction of acceptors when they suffer the attack 198 of drought; in other words, it is the possibility of potential loss caused by drought. Adaptive 199 capacity is the ability of human being to defend and mitigate the drought disasters. Vulnerability 200 evaluation model is as follows:

Vulnerability index= Exposure index\*Sensitivity index

Adaptive capacity index

Many factors can affect the vulnerability of drought, for example, population, Gross Domestic 202 203 Product, revenue, water resource, sown area, irrigation area and so on. Due to the interaction of 204 multiple factors, the vulnerability of drought in different counties may be very different. The 205 vulnerability evaluation factors should be chosen from multiple impacts factors, which could 206 precisely reveal the characteristics of vulnerability of drought. In this research, 15 factors were 207 selected (table 2), including permanent residents, population density, education level of population, aging rate and so on. The drought trends may change under climate change, which 208 209 would have an impact on hazard evaluation of drought. But in regard to vulnerability, its trends 210 may be much more complicated in the future, because the vulnerability of drought involves many different aspects, such as human being, economy, environment, society, eco-system and so on. 211 212 The uncertainty would be very high if the vulnerability factors of drought in the future are predicted. So the recent situations of vulnerability factors of drought are usually treated as the 213 typical situation for future vulnerability evaluation. Based on the available data, the vulnerability 214 215 factors of drought in 2012 are selected and it is hypothesized that the vulnerability in the future (from 2016 to 2050) is the same as the vulnerability in 2012. 216

### Tab. 2 Vulnerability evaluation index system of Drought

| Indexes     | Factors                          | Introduction                                           |  |
|-------------|----------------------------------|--------------------------------------------------------|--|
|             | Pormanant residents              | The population who live in one county for more than 6  |  |
|             | remanent residents               | months per year                                        |  |
| <b>F</b>    |                                  | The monetary value of all the finished goods and       |  |
| Exposure    | Gross Domestic Product           | services produced per year within a county's borders   |  |
|             |                                  | The area of crops which are planted per year in one    |  |
|             | sown area of crops               | county                                                 |  |
|             | Population density               | The density of population in one county                |  |
|             |                                  | The percentage of residents with inadequate education  |  |
|             | Education level of population    | which are under college level in the whole population  |  |
|             |                                  | which is beyond 6 years old                            |  |
|             | Aging rate                       | The percentage of population which is between 20       |  |
| Soncitivity |                                  | years old and 60 years old in the whole population     |  |
| Sensitivity |                                  | The percentage of agricultural population in the       |  |
|             | orbanization rate                | permanent residents of one county                      |  |
|             |                                  | The volume of water resources per year for one person  |  |
|             | Per capita water resources       | in one county                                          |  |
|             | Water consumption for 10000 Yuan | The volume of water consumption per year for 10000     |  |
|             | of GDP                           | Yuan of GDP in one county                              |  |
|             | 1 C                              | The whole fiscal revenue of local government per year  |  |
|             | Local fiscal revenue             | in one county                                          |  |
|             |                                  | The percentage of irrigated area per year in all the   |  |
| :           | Effective irrigated areas        | arable land of one county                              |  |
| Adaptive    |                                  | The volume of water supply per year by the pipes in    |  |
| capacity    | water supply capacity            | one county                                             |  |
|             | Water storage capacity           | The volume of water storage per year in one county     |  |
|             | Per capita income                | The average income of residents per year in one county |  |
|             | Forest coverage rate             | The percentage of forest coverage in one county        |  |

### 218 2.3.1 Exposure index

Exposure index is an important part of vulnerability evaluation index system, which reveals the 220 extent, quantity and size of acceptors. In one county, human being is usually considered as the 221 first key element, because people-oriented is the core of risk assessment. When drought 222 disasters break out, the survival of human being should be the first place primacy. Then 223 agriculture is considered as another important factor, which is related to the reduction of yield or even total crop failure. In addition, economy is thought to be an indispensable factor, which 224 225 refers to the potential losses due to drought disasters. In this research, Permanent residents (PR), 226 Gross Domestic Product (GDP) and Sown area of crops (SRC) are selected to be exposure factors 227 of drought.

Fig.1 The percentile of PR (Permanent residents), GDP (Gross Domestic Product) and SRC (Sown area of crops) in
 2373 counties of whole China

Exposure index (EI) model is established based on the above factors. First, the three factors should be converted into non-dimensional. The percentile distributions of PR, GDP and SRC are shown in Fig.1. Due to the huge gaps of different counties, most numerical values of the three exposure factors are smaller than 1.5, especially in which most numerical values of GDP factor are smaller than 1.25. Then the exposure indexes in 2373 counties of whole China are calculated according to the following model.

### $EI=\sqrt[3]{PR * GDP * SRC}$

Based on the calculation and percentile distribution, the exposure indexes in 2373 counties are classified into five levels, including highest exposure, high exposure, moderate exposure, low exposure and lowest exposure. Almost 268 counties are under highest exposure; 495 counties are under high exposure; 777 counties are under moderate exposure; 452 counties are under low exposure; 381 counties are under lowest exposure.

### 243 **2.3.2 Sensitivity**

Sensitivity index is the core of vulnerability evaluation index system, which reveals the vulnerable levels of acceptors when they are faced with different stressors (such as extreme events, disasters, human activities and so on). Different stressors may cause different sensitivity indexes for the same acceptor. For example, when crops suffer a serious flood, the main sensitivity indexes may be the flood resistance of crops; but when crops suffer a severe drought, the main sensitivity indexes may be the drought resistance of crops. So it is important to choose the appropriate factors of sensitivity index for drought. In this research, six factors are selected to

- reveal the sensitivity of drought in different counties, including Population density (PD),
- Education level of population (ELP), Aging rate (AR), Urbanization rate (UR), Per capita water
- resources (PWR) and Water consumption for 10000 Yuan of GDP (WCG).

Fig.2 The percentile of PD (Population density), ELP (Education level of population), AR (Aging rate), UR
 (Urbanization rate), PWR (Per capita water resources) and WCG (Water consumption for 10000 Yuan of GDP) in
 2373 counties of whole China
 As the same with exposure index, the six factors should be converted into non-dimensional.
 The percentile distributions of PD, ELP, AR, UR, PWR and WCG are shown in Fig.2, in which the
 differences among numerical ranges of the six sensitivity factors are revealed. Most numerical

values of ELP and UR are bigger than 2.0; most numerical values of PD, PWR and WCG are smaller
 than 1.5; most numerical values of AR are concentrated in 2.0 ~ 2.5. It is important to reduce the
 magnitude gaps of different factors for sensitivity evaluation. The sensitivity indexes (SI) in 2373
 counties of whole China are calculated according to the following model.

$SI = \sqrt[6]{PD * ELP * AR * UR * PWR * WCG}$ 266 Based on the calculation and percentile distribution, the sensitivity indexes in 2373 counties 267 are classified into five levels, including highest sensitivity, high sensitivity, moderate sensitivity,

are classified into five levels, including fighest sensitivity, figh sensitivity, inductate sensitivity,
 low sensitivity and lowest sensitivity. Almost 340 counties are under highest sensitivity;
 counties are under high sensitivity;
 803 counties are under moderate sensitivity;
 463 counties are
 under low sensitivity;
 325 counties are under lowest sensitivity.

# 271 2.3.3 Adaptive capacity

Adaptive capacity is the opposite of vulnerability, which reveals the capacity of suffering and 273 defending the stressors. When adaptive capacity is much stronger, the vulnerability will be much 274 lower. Usually adaptive capacity index is connected closely with sensitivity index. For example, 275 education level of population factor could reveal the sensitivity of population with different levels 276 of education, which could also indicate the degrees of adaptive capacity in different counties. In 277 other words, high education level of population in one county is much bigger, the sensitivity may 278 be lower and the adaptive capacity may be higher. So it is necessary to distinguish the sensitivity 279 factors and adaptive capacity factors. In this research, six factors are selected to reveal the 280 adaptive capacity of drought in different counties, including Local fiscal revenue (LFR), Effective 281 irrigated areas (EIA), Water supply capacity (WPC), Water storage capacity (WSC), Per capita 282 income (PCI) and Forest coverage rate (FCR).

Fig.3 The percentile of LFR (Local fiscal revenue), EIA (Effective irrigated areas), WSC (Water storage capacity), 285 WPC (Water supply capacity), PCI (Per capita income) and FCR (Forest coverage rate) in 2373 counties of whole 286 China

As the same with exposure index and sensitivity index, the six factors of adaptive capacity 288 should be converted into non-dimensional. The percentile distributions of LFR, EIA, PCI, WPC, WSC and FCR are shown in Fig.3, in which the differences among numerical ranges of the six 289 290 adaptive capacity factors are revealed. The distribution ranges of WSC and FCR are bigger than 291 the other four factors, which cover a wide range of 1.0~5.5. Most numerical values of EIA and PCI 292 are centralized in 2.0~3.5; most numerical values of WPC are from 1.0 to 2.0; most numerical 293 values of LFR are smaller than 0.5. Based on exposure index model and adaptive capacity index 294 model, the adaptive capacity indexes (ACI) in 2373 counties of whole China are calculated 295 according to the following model.

## 296

### $ACI= \sqrt[6]{LFR * EIA * PCI * WPC * WSC * FCR}$

According to the calculation and percentile distribution, the adaptive capacity indexes in 2373 298 counties are classified into five levels, including highest level, high level, moderate level, low level 299 and lowest le