# Peer review of "Risk assessment of meteorological drought in China under RCP"

_Natural Hazards and Earth System Sciences, 2016_

## Referee Comment (RC1) · Anonymous Referee #1 · 12 Jan 2017

This paper addresses a relevant scientific question within the scope of NHESS. The questions tackled are interesting and deserve to be studied, however the article has many structural problems. I will mention the most glaring ones, but there are more.

The first problem is that it is not possible to know if the methods used by the authors were developed by themselves or not. The authors do not sufficiently nor correctly cite the literature in order to put their work into context. When they use new methods, if they use new methods at all, they do not explain what is new, why it is new and how did they develop the method. They do not justify the choices made. That is enough to reject the paper.

Furthermore, the methodology is weak and it is not developed enough. The authors

make use of a single climate simulation, but they do not discuss how this limitation impacts the uncertainty of their results. The simulations is downscaled to a higher resolution, but they do not discuss how this adds uncertainty to the study. We do not know if the model used is able to simulate droughts in the present. They should validate the capacity of the model to reproduce drought patterns in China. Then they develop an integrated index of drought hazard, but their choices seem quite arbitrary. They don't justify the weights used, they do not explain why they use SPI-12 and not another scale, they do not even sufficiently explain how they calculate the SPI (I guess they use a gamma distribution, but they do not explain this). Afterwards they estimate the vulnerability. They use a lot of data in order to estimate vulnerability, which is good, but they do not explain where the data comes from, how do they determine which factors are related to exposure, sensitivity or adaptive capacity. They do not justify if the factors are independent enough from each other. They do not test the robustness of the index (how does it change if we change a the factors?). They do not explain how they normalize the data, etc. They do not discuss the quality of the statistical county data used. The list is, unfortunately, long.

The figures do not have enough quality and there are not enough figures. The box plot figures are not well explained. The vertical axis is labeled "Percentiles", when it does not show percentiles. The data shown is dimensionless and thus the axis should not be labeled. Maybe "Percentiles" should be in the title of the plot. The box plots show the percentiles, but the authors do not explain what the whiskers and the limits of the box mean (which percentiles do they represent?). The maps in Figure 4 show the risk in the hazard in the future, but the authors say they show trends. These are not trends, these are maps of the hazard in a future time frame. The authors should also show the values of the hazard in the present climate and maps showing the differences. Same happens for the risk.

Writing an article in English is not easy when your mother tongue is very different to English. I have the same problem and it is often painful. But sometimes it is not clear

if the authors have difficulties with the language, which they have, or the underlying scientific concepts. The author's should send the article to a native English speaker in order to correct the grammar and the vocabulary.

My point of view is that the article should be rejected. However, I encourage the authors to continue their work and re-submit the article in the future when their work is mature enough. The topic of the article is interesting and I am sure that studies on the future drought risk in China are necessary. Furthermore, the statistical data the authors use in order to calculate the vulnerability is very interesting. Unfortunately they submitted their paper too early.

---

## Referee Comment (RC2) · Anonymous Referee #2 · 20 Jan 2017

**General comments**

This manuscript performs an extensive assessment of the progression of risk linked to the evolution of droughts in different climate change scenarios in China from 2016 to 2050. The paper has remarkable scientific novelty and the topic meets the scope of NHESS. I would like to highlight the interest of the exhaustive compilation of factors linked to the computation of risk in China, the thorough spatial data coverage, the careful analysis of the evolution of the index of drought hazard for the next decades, and the estimation of risk change until 2050 in different climate change scenarios.

However, the paper has many problems and, as a result, its message lacks the robustness needed for acceptation. Although I recommend rejecting the paper, I would advise the authors to keep on ameliorating the document and try a re-submission when the existing issues are fully resolved.

**Specific comments**

In the following lines I will present, in order of appearance, some of the main problems,

- The overall level of the English used needs improvement because the paper has many grammatical and syntax errors that often affect the clarity of the message conveyed. Thus, it would be good if a native speaker could revise and correct the manuscript.

- **Lines 27-29.** Repeating the exact same sentence in the abstract and other parts of the document is not very palatable from a reader point of view. Please avoid this practice.

- **Line 57.** I am wondering which are these 'human activities' that mainly affect the socio-economical drought. Please add the references accordingly.

- **Lines 66-68.** In these lines various drought indexes are stated to be better depending on each drought situation. Each allocation should be justified and include the corresponding references.

- **Lines 119-121.** Why do you use HadGEM2-ES? Please, explain your decision (adding references if needed).

- **Lines 144-150.** This paragraph lacks references.

- **Line 153.** '. . .fitted to a probability distribution'. Which distribution? Which are the more common? Is the SPI computation method suited for any kind of distribution? Although the reasons for choosing the SPI can be drawn from what you say in this paragraph, please state them more clearly.

- **Line 159.** '. . .the SPI Values table'. After 'table' should be the explicit reference to the table.

- **Line 160.** From the table I would guess that a meteorological drought event would be triggered when the SPI is below -0.5 (not -1.0).

- **Line 168.** The origin of the weighting factors is not explained. Why are they 0.1 or 0.4 and not 0.03 and 0.12?

- **Line 169-170.** The shape parameter and scale parameter of which distribution?

- **Line 178-180.** Have you developed the Integrated Index of Drought Hazard (IIDH)? Where it comes from? Please add references or state clearly the novelty of the method. In case it is an index developed by you then explain it comprehensively.

- **Lines 195-196.** 'Most of researchers have accepted the vulnerability evaluation model'. Add references to this statement.

- **Line 201.** Where does the vulnerability model come from? Please add references and explain the position of the different parameters in the mathematical expression.

- **Lines 202-204.** These statements need references to support them.

- **Lines 206-216.** Why 15 and not 20 or 10? Could you explain the process of selection? How do you rank them? How do you verify their performance? How

do you verify that are independent of each other? On what aspects do you rely to take your decisions? Please explain and add references when necessary.

- **Line 227.** The figure is too small and the 'y' label is not correct.

- **Line 237.** Please justify the inclusion of the different parameters in the expression and, also, the structure of the expression.

- **Line 240-242.** These data would be more readable if it was summarised in a table.

- **Lines 250-253.** Why do you choose six parameters? Please explain the reasons. Add references when necessary.

- **Line 255.** The figure is too small and the 'y' label is not correct.

- **Line 265.** Why does the model have this structure? Explain.

- **Lines 268-270.** These data would be more readable if it was summarised in a table.

- **Lines 274-282.** Please add references to the assertions made in this paragraph.

- **Line 284.** The figure is too small and the 'y' label is not correct.

- **- Line 296.** Justify the inclusion of the different parameters in the expression and, also, the structure of the expression.

- **Line 297-301.** These data would be more readable if it was summarised in a table.

- **Lines 304-306.** This statement needs reference support.
- **Lines 316-317.** Justify why do you choose these scenarios. Are there only those?

- **Lines 345-361.** Please summarise all these data in a table.

- **Lines 362-373.** These lines are a bit cumbersome. Please simplify and clarify the text.

- **Line 385.** In this caption there is an excess of one parenthesis: '((picture c))'

- **Lines 417-431.** Please summarise all these data into a table.

- **Line 434.** 'The disasters become much more serious'. Please add a reference for this statement.

- **Line 437-438.** 'In summary, there are several conclusions of this study which need to b discussed'. This line can be removed.

- **Line 449.** Which are these 'more threatened' regions?

- **Lines 465-466.** Source of this statement?

- **Lines 475-476.** 'The 15 factors are independent. . .' How do you know? Explain accordingly in the methodology section.

---

## Author Comment (AC1) · 3 Apr 2017

We think the rejection is arbitrary and irrational. The methods used in this article are developed by ourselves. We have cited proper literatures which are put into the right place. The whole article is interpreting the process of risk assessment of meteorological drought in China under RCP scenarios from 2016 to 2050. Risk assessment is the core of the article, in which the climate simulation is just the background. HadGEM2-ES is more proper than the other GCMs in East Asia. Many studies have been done using HadGEM in China. So we choose it for the risk assessment. The reason of using SPI-12 is explained in Page 4, Line 141-155. The weighting factors are based on experts experience; they are scored by 20 experts who are familiar with SPI and drought.

The data of vulnerability evaluation comes from Chinese statistical yearbooks. The reason of choosing these vulnerability evaluation indexes is revealed in "2.3 Vulnerability evaluation". The figures have enough quality to reveal the distribution of hazard, vulnerability and risk under RCPs. The box plot figures are modified and explained. The maps in Figure 4 show the hazards in the future. It is not necessary to show the present hazard and risk in this article. Although your rejection is unreasonable, we also thank you for your reviewing.

---

## Author Comment (AC2) · 3 Apr 2017

Thank you for your detailed and constructive suggestions. We have modified the article according to the suggestions item by item.

Line 27-29. The repeated sentence is removed.

Line 57. When meteorological drought happens, people should take actions to fight against it through water supply network, water resource management, reservoirs and so on; if the actions are not enough or even wrong, socio-economical drought probably happens. The references are cited in the article. (Wihite, 2000; Wisser et al, 2010)

Lines 66-68. The statement is our own opinion. We summarized it from the definitions

of different drought indexes.

Lines 119-121. HadGEM2-ES is more proper than the other GCMs in East Asia. Many studies have been done using HadGEM in China. (Xu Yinlong et al, 2006; Xiong Wei et al, 2009; Jiang Ying et al, 2010; Yang Honglong et al, 2010)

Lines 144-150. We add one reference in this paragraph. (Hong Wu et al, 2005)

Line 153. In this article, the probability distribution of 12-month SPI is calculated. It is transformed into a normal distribution in order to identify wetter and drier climates situations.

Line 159. The reference is added in the table. (Guttman, 1998)

Line 160. Yes. A meteorological drought event would be triggered when the SPI is below -0.5.

Line 168. The weighting factors are based on experts experience. The weighting values are scored by 20 experts who are familiar with SPI and drought.

Line 169-170. The shape parameter and scale parameter of precipitation distribution are estimated.

Lines 178-180. We have developed the Integrated Index of Drought Hazard. It is based on SPI. According to different weighting factors and the frequencies of SPI, the integrated indexes are developed to evaluate drought hazard. The method is as follows: IIDH=0.1*F1+0.2*F2+0.3*F3+0.4*F4 IIDH is the Integrated Index of Drought Hazard; F1 is the frequency of mild drought; F2 is the frequency of moderate drought; F3 is the frequency of heavy drought; F4 is the frequency of excessive drought.

Lines 195-196. A reference is added. (Houghton et al., 2001)

Line 201. The vulnerability model comes from IPCC report (IPCC, 2007). The reference is added.

Lines 202-204. These statement is our own understanding of vulnerability according to the previous studies all over the world.

Lines 206-216. The selection of vulnerability factors is mainly based on experts experience and it is restricted by the credibility and completeness of data and the independent of factors. The 15 factors are not ranked. The independent of factors is considered before we choose the factors. From the introduction of factors in Table 2, it is clear that we just choose one attribute of each factor. So the interaction of selected factors is avoided as much as possible.

Line 227. The figure is enlarged and the "Y" label is removed.

Line 237. The inclusion of the different parameters is stated from Line 228 to 230. The structure of the expression would reveal the level of exposure index in different counties according to 3 factors(PR, GDP, SRC).

Line 240-242. These data is summarized in a table.

Line 255. The figure is enlarged and the "Y" label is removed.

Line 265. The structure of the expression would reveal the level of sensitivity index in different counties according to 6 factors(PD, ELP, AR, UR, PWR and WCG).

Line 268-270. These data is summarized in a table.

Line 274-282. The statement is summarized by ourselves.

Line 284. The figure is enlarged and the "Y" label is removed.

Line 296. The structure of the expression would reveal the level of adaptive capacity index in different counties according to 6 factors(LFR, EIA, PCI, WPC, WSC and FCR).
Line 297-301. These data is summarized in a table.

Line 304-306. The reference is added. (IPCC, 2007)

Line 316-317. In this article, RCP (2.6, 4.5, 6.0, 8.5) scenarios are chosen in order to

compare the different risks under different RCPs in the future.

Line 345-361. These data is summarized in a table.

Line 362-373. These sentences are simplified and clarified.

Line 385. The redundant parenthesis is removed.

Line 417-431. These data is summarized in a table.

Line 434. The reference is added.

Line 437-438. The sentence is removed.

Line 449. It is about the most risky regions threatened by meteorological drought in the future.

Lines 465-466. It comes from the China National Commission for Disaster Reduction.

Lines 475-476. In the methodology section we have explained it.